# NoTeNet: Normalized Mutual Information-Driven Tuning-free Dynamic Dependence Network Inference Method for Multimodal Data

## Abstract

Dynamic Dependence Network (DDN) inference is crucial for understanding evolving relationships in multimodal time series web data, with broad applications in fields like medical and financial network analysis. The inherent dynamic nature, temporal continuity, and heterogeneous data sources in multimodal time series data pose three fundamental challenges: computational efficiency, prediction stability and robustness, and modality quality disparity. Previous methods, generally lacking utilization of multiple modalities, either struggle with computational efficiency due to the time-intensive manual hyperparameter tuning, or compromise prediction stability and robustness by neglecting temporal coherence. To address these challenges, we propose a Normalized mutual information-driven Tuning-free Dynamic Dependence Network inference method for multimodal data, namely NoTeNet. NoTeNet provides a promising paradigm that can integrate two different data modalities to enhance prediction accuracy. It uses normalized mutual information transforms noisy auxiliary data into relationship matrices and employs a kernel function for smooth temporal estimation. Additionally, NoTeNet significantly reduces the need for manual hyperparameter adjustments, offering a tuning-free approach with theoretical guarantees. On various synthetic datasets and real-world data, NoTeNet demonstrates superior prediction accuracy and efficiency without the need for hyperparameter tuning, making it potential for a wide range of web data applications.

## CCS Concepts

• **Computing methodologies → Learning in probabilistic graphical models**; • **Networks** → *Network structure*.

## Keywords

Dynamic Dependence Network, Multimodal Fusion, Web Time Series Data

**ACM Reference Format:**

Anonymous Author(s). 2024. NoTeNet: Normalized Mutual Information-Driven Tuning-free Dynamic Dependence Network Inference Method for Multimodal Data. In *Proceedings of Make sure to enter the correct conference title from your rights confirmation email (The Web Conference)*. ACM, New York, NY, USA, 12 pages. https://doi.org/XXXXXXX.XXXXXXX

## 1 Introduction

Dynamic Dependence Network (DDN) inference is a pivotal task in web data analysis, emphasizing the study of evolving relationships between entities over time. By analyzing temporal dependencies, DNN offers insights into evolving interactions, which are vital for the analysis and monitoring of diverse web systems, including finance, medical networks, and social platforms. For instance, DDN inference is applied to functional Magnetic Resonance Imaging (fMRI) data to predict functional connectivity networks in the brain for neurological and psychiatric disorder diagnosiss [43]. Predominantly dependent on (fMRI), the prediction cannot accurately capture the brain's rapid dynamic shifts because of fMRI's slow sampling rate [15]. With technological advancements, incorporating brain data from modalities like Electroencephalography (EEG) has become a promising strategy to enhance prediction. However, Electroencephalography (EEG), despite its high temporal resolution, has been notably underutilized, a situation largely attributable to the difficulties in integrating data from different modalities [35]. A similar situation occurs in stock-news data analysis for financial network, where stock data alone cannot capture external events or market sentiment [19, 27]. Despite the difficulty, their multimodal integration presents a highly potential avenue for advancing DDN research [22, 46].

There is a range of methodologies [5, 21, 23] that employ precision matrix estimation to predict dynamic dependence networks. These methods utilize the inverse of the covariance matrix to highlight the conditional independence among different entities, thereby offering a more precise understanding of the interaction network. However, these methods still encounter three challenges in the process of multimodal web data fusion and inference:

Firstly, **Computational efficiency**. In dynamic network prediction, frequent estimation of networks across multiple time points generates a large computational burden, especially when real-time data is continuously updated. Relying on manual parameter adjustments for each time point, such as selecting regularization parameters, becomes impractical under these conditions. Most of the previous works [2, 24] in precision matrix estimation typically rely on the selection of an appropriate regularization parameter value to achieve optimal performance. However, setting the level of regularization requires computationally intensive methods like cross-validation, thereby compounding the challenge. Consequently, it is essential to develop an estimator that can achieve optimal performance without any manual parameter adjustments.

Secondly, **Prediction stability and robustness**. In web time series data, network structures across adjacent timestamps often exhibit strong similarity and continuity in a period. For instance, during continuous music listening, brain functional connectivity in auditory regions between adjacent timestamps remains highly similar, reflecting the uninterrupted nature of the stimulus [17].

Previous methods [11, 16], which assume temporal independence, typically estimate precision matrices separately for each timestamp. This practice can overlook temporal coherence, where similar patterns across adjacent time points may exist, and accounting for these could improve prediction stability and robustness.

Finally, **Modality quality disparity**. In real-world settings, the quality of different modalities usually varies due to unexpected environmental factors or sensor issues. fMRI, classified as the targeted modality, serves as the primary variable of interest. These datasets typically follow a direct temporal sequence and are the core focus for prediction or analysis. They tend to have higher accuracy, lower noise levels, and greater reliability and are generally assumed to follow a sub-Gaussian distribution. EEG, as the auxiliary modality, provides supplementary information that enriches the analysis, albeit with higher noise and no clearly defined distribution [1]. The lower data quality of the auxiliary modality compared to the targeted modality can lead to unreliable multimodal fusion outcomes. Therefore, it is desirable to develop a method capable of effectively processing and integrating information from both modalities, despite their substantial differences in their noise characteristics, data distributions, and other inherent properties [37].

To address the above challenges, we propose NoTeNet, a Normalized mutual information-driven Tuning-free Dynamic Dependence Network inference method for multimodal data. In the first stage, we introduce the normalized mutual information to transform the auxiliary dataset into the relationship matrices, aligning it temporally with the samples from the targeted dataset. As mentioned, auxiliary data are noisy and follow unknown distributions, in which traditional measures like Pearson correlation fail to capture non-linear dependencies. As a robust alternative, normalized mutual information does not assume a specific data distribution. By the normalizing step, it makes the measure less sensitive to large entropy differences and ensures interpretability between 0 and 1, which enhances its robustness to noise.

In the second stage, Instead of the temporal independency assumption, we take full advantage of the data from adjacent timestamps by using a kernel function to ensure the temporal coherence. To lower the huge tuning computational cost, our method greatly simplifies the tuning procedure, verifying the tuning-free property with a theoretical guarantee.

Overall, our contribution can be summarized as follows:

- Novel DDN paradigm for multimodal data: We introduce an innovative DDN inference designed to exploit the underlying time-varying graph structure with multimodal data fusion.
- Tuning-free method: The penalty level of NoTeNet is automatically set to achieve the optimal convergence rate for the estimation of each column of the precision matrices.
- Theoretical guarantee: We detailedly study the theoretical properties of the proposed estimator. We guarantee the estimation consistency and convergence rate of our method and verify its tuning-free properties.
- Experimental evaluation: On multiple synthetic datasets, NoTeNet outperforms the other baselines in prediction accuracy without the need for hyperparameter tuning. We

also implement our method on the real-world datasets and demonstrate the efficiency of NoTeNet.

## 2 Related Work

### 2.1 Dynamic Dependence Network

Dynamic Dependence Network (DDN) [4, 48] is a model used to capture and analyze time-varying relationships among multiple entities within a network. Unlike static networks, which assume fixed connections over time, DDNs allow for the dynamic adaptation of connections, reflecting how dependencies between entities evolve. This makes DDNs particularly suitable for analyzing web data, where interactions and relationships between entities can change rapidly over time.

Statistical learning methods [12, 40, 47] for DDN estimation provide several advantages, including interpretability, theoretical guarantees, and a lightweight nature. They are computationally efficient, requiring fewer parameters and less training data, making them suitable for real-time or resource-constrained environments. Such approaches can be categories into time series models [32] and graphical models [42].

*Time series models.* Traditional time series models, such as vector autoregression (VAR) and state-space models, are often used to capture time-dependent interactions. These models excel in identifying linear relationships between entities over time and are particularly useful in simpler dynamic networks. However, they tend to be less effective when dealing with high-dimensional datasets [33] and complex interaction structures, which are typical in many web data applications [48].

*Graphical Models.* Precision matrix estimation-based graphical models, typically assuming Gaussian or sub-Gaussian distributions, have become a central focus for DDN estimation within statistical learning. These include methods such as time-varying graphical lasso, dynamic Gaussian graphical models (DGGM) [34], and regularized precision matrix estimation techniques like time-varying CLIME [3, 44]. These approaches are particularly well-suited for high-dimensional data, as they allow for learning sparse dependency structures that evolve over time.

### 2.2 Normalized Mutual Information

*Mutual information.* Mutual Information (MI) is a fundamental concept in information theory, used to measure the dependency between two random variables. Unlike metrics such as Pearson correlation, Spearman's rank correlation, or cosine similarity, which often assume specific types of relationships (e.g., linear for Pearson) or data distributions, MI is non-parametric and does not require any assumptions about the underlying data distribution. This makes MI ideal for capturing both linear and non-linear dependencies across diverse variables [10, 25]. For two random discrete variables $X$ and $Y$, the MI is defined as:

$$\text{MI}(X; Y) = \sum_{x \in X} \sum_{y \in Y} P(x, y) \log \left( \frac{P(x, y)}{P(x)P(y)} \right) \quad (1)$$

where $P(x)$ is the probability of the variable $X$ taking a specific value $x$. In practice, $P(x)$ is estimated based on the frequency of occurrences of $x$ when $X$ is a discrete variable. When $X$ and $Y$ are

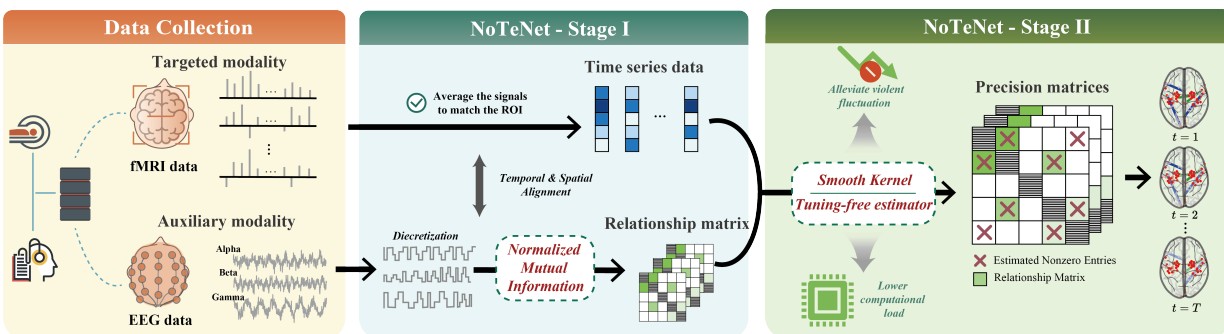

**Figure 1: The pipeline of the proposed method. Using fMRI-EEG data as an example, the process begins with the data collection phase, where EEG and fMRI data are gathered from the Internet of Things. In the first stage of NoTeNet, time series data from the targeted modality for each ROI are extracted and the EEG data are discretized for denoising. Then we introduce normalized mutual information to obtain the relationship matrices. The second stage contains two key operations: a) We utilize the smooth kernel function to ensure the temporal coherence of the estimation. b) We leverage the relationship matrices to enhance the precision matrix estimation with a tuning-free technique.**

independent, MI equals zero; when there is any form of dependency, MI becomes positive.

*Normalized mutual information.* Although MI is useful, it is sensitive to noise and lacks a clear upper bound, which makes it harder to interpret and compare. Nomarlized MI restricts the MI value to $[0, 1]$ range, mitigating the impact of extreme noise and providing more interpretability through normalization [29].The formula for discrete variables is:

$$\text{NMI}(X; Y) = \frac{\text{MI}(X; Y)}{\frac{1}{2}(H(X) + H(Y))} \quad (2)$$

$H(X) = -\sum_x P(x) \log P(x)$ and $H(Y)$ represent the entropy of the variables $X$ and $Y$, respectively.

In this paper, while many auxiliary modalities–such as EEG data– are continuous in nature, we discretize them to reduce the effect of noise. By segmenting continuous data into bins, we smooth out random fluctuations and make the data more robust against noise interference.

## 2.3 Neighborhood Approach for Precision Matrix Estimation

The task of precision matrix estimation involves deducing the inverse covariance matrix for a multivariate entity. This matrix is pivotal in uncovering the conditional independence among variables, serving an array of applications from learning graphical models as noted [31, 41]. A well-known approach for estimating the precision matrix is the graphical lasso [9], a penalized maximum likelihood estimator:

$$\hat{\Omega} = \underset{\Omega \succ 0}{\arg\min} \; -\log\det(\Omega) + <\Omega, \Sigma> + \lambda\|\Omega\|_1, \quad (3)$$

where $\Omega$ must be symmetric positive definite and the penalty parameter $\lambda \geq 0$ and the covariance matrix is $\hat{\Sigma} = \frac{1}{n}\sum_{i=1}^n \mathbf{X}_i\mathbf{X}_i^\top$. Based on (3), [26] proposed the neighborhood selection estimate

with Lasso:

$$\hat{\boldsymbol{\beta}}_j = \underset{\boldsymbol{\beta}_j : \beta_{jj}=0}{\arg\min} \left( \|\mathbf{X}_{:j} - \mathbf{X}\boldsymbol{\beta}_j\|_2^2 + \lambda\|\boldsymbol{\beta}_j\|_1 \right), \quad (4)$$

where $\mathbf{X}_{:j}$ denotes the $j$-th column, e.g, the $j$-th feature, of $\mathbf{X}$, $\beta_{kj}$ denotes the $k$-th element of $\boldsymbol{\beta}_j$. The elements of $\boldsymbol{\beta}_j$ are actually determined by the precision matrix that $\beta_{kj} = -\Omega_{kj}/\Omega_{jj}$. Neighborhood selection estimates the conditional independence of each feature separately in order to effectively estimate the structural zeros of the precision matrix.

## 3 The Proposed Method

### 3.1 Overview

As depicted in Figure. 1, after data collection, our pipeline for DDN prediction task can be divided into two main stages: 1) The first stage focuses on the processing of data across two disparate datasets from different modalities. In scenarios with two datasets, we designate one as the auxiliary dataset and the other as the targeted dataset. This stage involves the transformation of the auxiliary dataset into a relationship matrix using normalized mutual information, followed by temporal alignment with the targeted dataset's samples. This alignment step ensures that the data from both sources are synchronized over time for improved integration in subsequent analysis. 2) The second stage (Section 3.2) is dedicated to using precision matrix estimation to predict the time-varying dependence network. Leveraging the relationship matrix derived from the auxiliary dataset, we integrate this information into the estimation of the precision matrix alongside the targeted time series data. This step includes the use of a kernel function to account for temporal continuity, ensuring smooth estimation over time. To simplify the process, we employ a tuning-free approach using scaled lasso, which eliminates the need for manual hyperparameter adjustments.

## 3.2 NoTeNet

*Notations.* We represent the time series data from the targeted dataset by $\mathbf{X} \in \mathbb{R}^{T \times p}$, where $T$ is the number of time points and $p$ is the number of features. Referring to the two examples provided above, we denote the data from the auxiliary dataset as $\mathcal{A} = \{\mathbf{A}^{(1)}, \mathbf{A}^{(2)}, \dots, \mathbf{A}^{(T)}\}$, with each $\mathbf{A}^{(t)} \in \mathbb{R}^{n \times p}$ representing the data at time $t$. For the targeted time series, $\mathbf{X}_{t,*} \in \mathbb{R}^p$ denotes the $t$-th sample, which we abbreviate as $\mathbf{X}_t$ for simplicity. The notation $\mathbf{A}_{*,i}^{(t)} \in \mathbb{R}^n$ specifies the $i$-th feature at the $t$-th timestamp in the auxiliary data. Our goal is to estimate a series of precision matrices $\{\hat{\Omega}^{(1)}, \hat{\Omega}^{(2)}, \dots, \hat{\Omega}^{(T)}\}$ from the combination of two modalities.

*Assumption.* Consider independent variables $\mathbf{X}_t$ distributed as $\mathcal{N}(0, \Sigma^{(t)})$. Each $\mathbf{X}_t$ is linked to a corresponding undirected graph $G(t)$, defined by the zero entries in the precision matrix $\Omega^{(t)}$. We operate under the assumption that the probability distribution, or the law, of $\mathbf{X}_t$ undergoes smooth variations.

To capture the complex relationships between entities from the auxiliary dataset, we utilize the normalized mutual information to estimate the relationship matrices:

$$\theta_{ij}^{(t)} = \text{NMI}\left(A_{*,i}^{(t)}, A_{*,j}^{(t)}\right) = \frac{\text{MI}\left(A_{*,i}^{(t)}; A_{*,j}^{(t)}\right)}{\frac{1}{2}\left(H\left(A_{*,i}^{(t)}\right) + H\left(A_{*,j}^{(t)}\right)^{(t)}\right)} \quad (5)$$

where

$$H\left(A_{*,i}^{(t)}\right) = -\sum_{a \in A_{*,i}^{(t)}} P(a) \log P(a) \quad (6)$$

and

$$\text{MI}\left(\mathbf{A}_{*,i}^{(t)}, \mathbf{A}_{*,j}^{(t)}\right) = \sum_{a \in A_{*,i}^{(t)}} \sum_{a' \in A_{*,j}^{(t)}} P(a, a') \log\left(\frac{P(a, a')}{P(a) \cdot P(a')}\right) \quad (7)$$

After obtaining the relationship matrices extracted from EEG, our next step is to integrate it with fMRI time series data $\mathbf{X}$ to estimate the precision matrices.

To estimate the precision matrix of the $t$-the time point, we first define the weighted matrix:

$$\mathbf{X}^{(t)} = \mathbf{W}^{(t)}\mathbf{X}, \ \mathbf{W}^{(t)} = \text{diag}\left(\sqrt{\omega_1^{(t)}}, \sqrt{\omega_2^{(t)}}, \dots, \sqrt{\omega_T^{(t)}}\right) \quad (8)$$

where $t = 1, 2, \dots, T$ and

$$\omega_s^{(t)} = \frac{K_h(\frac{|s-t|}{T})}{\sum\limits_s K_h(\frac{|s-t|}{T})} \quad (9)$$

and $K(\cdot) : \mathbb{R} \mapsto \mathbb{R}$ is symmetric nonnegative kernel function and $K_h(\cdot) = K(\cdot/h)$. The selection of $K(\cdot)$ will be discussed later. It is noticed that this kernel function is closely related to the distance between two timestamps. The kernel function assigns samples closer to the current moment in sampling time greater weights to guarantee a stronger similarity between adjacent timestamps.

According to $\mathbf{X}_i \sim \mathcal{N}(0, \Sigma^{(i)})$, we have $\mathbf{X}_i^{(t)} \sim \mathcal{N}(0, \tilde{\Sigma}^{(i)} = \omega_i^{(t)}\Sigma^{(i)})$, represented by $\mathbf{X}_i^{(t)} = (X_{i,1}^{(t)}, X_{i,2}^{(t)}, \dots, X_{i,p}^{(t)}), i = 1, 2, \dots, T$. Then we have the following distribution $X_{i,j}^{(t)} | X_{i,-j}^{(t)} \sim \mathcal{N}_{p-1}(\tilde{\Sigma}_{j,-j}^{(t)}$

$[\tilde{\Sigma}_{-j,-j}^{(t)}]^{-1} X_{i,-j}^{(t)}, \tilde{\Sigma}_{j,j}^{(t)} - \tilde{\Sigma}_{j,-j}^{(t)}[\tilde{\Sigma}_{-j,-j}^{(t)}]^{-1}\tilde{\Sigma}_{-j,j}^{(t)})$, which is equivalent to the linear model:

$$X_{i,j}^{(t)} = \sum_{k \neq j} \beta_{kj}^{(t)} X_{i,k}^{(t)} + \epsilon_{ij}^{(t)}, \quad (10)$$

where $\epsilon_{ij}^{(t)} \sim \sigma_j^2(t) = \tilde{\Sigma}_{j,j}^{(t)} - \tilde{\Sigma}_{j,-j}^{(t)}[\tilde{\Sigma}_{-j,-j}^{(t)}]^{-1}\tilde{\Sigma}_{-j,j}^{(t)}$ is the error standard deviation, $\beta_{kj}^{(t)}$ is the regression coefficient, and $k = 1, 2, \dots, p$. In the regression approach to estimating sparse precision matrices, the elements of the precision matrix are mapped to regression coefficients and error variances through the following relationships:

$$\Omega_{kj}^{(t)} = -\frac{\beta_{kj}^{(t)}}{\sigma_j^2(t)}, \ \Omega_{jj}^{(t)} = \frac{1}{\sigma_j^2(t)}, \text{ for } 1 \leq k \neq j \leq p. \quad (11)$$

Therefore, we can estimate the precision matrices $\{\hat{\Omega}^{(1)}, \dots, \hat{\Omega}^{(T)}\}$ by solving a series of corresponding regression problems (10). To end it, we utilize the Scaled Lasso to estimate the regression coefficients $\beta_{kj}^{(t)}$ and the error variances $\epsilon_{ij}^{(t)}$.

Inspired by [3], we can solve the scaled lasso problem column by column. To be specific, we use $\mathbf{B}^{(t)} = (\beta_{kj}^{(t)})_{1 \leq k,j \leq p}$ to represent the matrix of the regression coefficients such that $\beta_{jj}^{(t)} = -1$ for $j = 1, \dots, p$. Let $\Lambda^{(t)} = \text{diag}\left(\sigma_1^{-2}(t), \dots, \sigma_p^{-2}(t)\right)$, the estimated precision matrix can be written as:

$$\Omega^{(t)} = -\mathbf{B}^{(t)}\Lambda^{(t)} = (-\sigma_1^{-2}(t)\boldsymbol{\beta}_1^{(t)}, \dots, -\sigma_p^{-2}(t)\boldsymbol{\beta}_p^{(t)}) \quad (12)$$

where $\boldsymbol{\beta}_j^{(t)} = \mathbf{B}_{*,j}^{(t)}$ is the $j$-th column of the matrix $\mathbf{B}^{(t)}$.

To estimate the targeted precision matrices $\{\hat{\Omega}^{(t)}\}_{1 \leq t \leq T}$, we propose the following estimator:

$$(\hat{\boldsymbol{\beta}}_j^{(t)}, \hat{\sigma}_j(t)) =$$

$$\underset{\sigma_j(t)>0, \boldsymbol{\beta}_j^{(t)}}{\arg \min} \frac{\boldsymbol{\beta}_j^{(t)\top}\hat{\Sigma}^{(t)}\boldsymbol{\beta}_j^{(t)}}{2\sigma_j(t)} + \frac{\sigma_j(t)}{2} + \lambda \sum_{k \neq j} \sqrt{\hat{\Sigma}_{kk}^{(t)}}|S(\theta_{kj}^{(t)}) \cdot \beta_{kj}^{(t)}|$$

$$(13)$$

where $\boldsymbol{\beta}_j^{(t)\top}\hat{\Sigma}^{(t)}\boldsymbol{\beta}_j^{(t)} = \|X_j^{(t)} - \sum\limits_{k \neq j} \beta_{kj}^{(t)}X_k^{(t)}\|_2^2/T$ and $\hat{\Sigma}^{(t)} = (\mathbf{X}^{(t)})^\top\mathbf{X}^{(t)}/T$, $S(z) = 1 - z$. Note that the relationship matrix is used in the regularization term to enhance the estimation.

Then we can get the estimated precision matrices according to (12):

$$\hat{\Omega}_0^{(t)} = -\hat{\mathbf{B}}^{(t)}\hat{\Lambda}^{(t)}, t = 1, 2, \dots, T. \quad (14)$$

The precision matrix is required to be symmetric as it represents the conditional dependency relationships between random variables within an undirected graph. However, (13) cannot guarantee the symmetry of the estimated precision matrices $\hat{\Omega}^{(t)}$. Therefore, we consider an additional symmetrization step:

$$\hat{\Omega}^{(t)} = \underset{\mathbf{M}:\mathbf{M}^\top=\mathbf{M}}{\arg \min} \|\mathbf{M} - \hat{\Omega}_0\|_1. \quad (15)$$

This optimization problem can solved by linear programming.

*Optimization Algorithm.* In this paper, we employ an iterative algorithm to address the solution of (13). To simplify the representation, we omitted the superscript of the symbols, e.g., $\hat{\mathbf{B}}^{(t)} \to \hat{\mathbf{B}}$. All the following operations are specific to the time point $t$. Here, $\hat{\mathbf{B}}(\lambda_0)$ represents the estimated $\hat{\mathbf{B}}$ with the hyperparameter $\lambda$. We can obtain the Lasso path by the estimation $\hat{\mathbf{B}}_{-j,j}(\lambda)$ satisfying the Karush-Kuhn-Tucker conditions:

$$\begin{cases} |S(\theta_{kj}^{(t)})|^{-1}\hat{\Sigma}_{kk}^{-1/2}\hat{\Sigma}_{k,*}\hat{\mathbf{B}}_{*,j}(\lambda) = -\lambda\,\mathrm{sgn}\left(\hat{\mathbf{B}}_{k,j}(\lambda)\right), & \hat{\mathbf{B}}_{k,j} \neq 0, \\ |S(\theta_{kj}^{(t)})|^{-1}\hat{\Sigma}_{kk}^{-1/2}\hat{\Sigma}_{k,*}\hat{\mathbf{B}}_{*,j}(\lambda) \in \lambda[-1,1], & \hat{\mathbf{B}}_{k,j} = 0, \end{cases}$$
$$(16)$$

for $k \neq j$, where $\mathrm{sgn}(\cdot)$ represents the sign functional. Here $\hat{\mathbf{B}}_{jj}(\lambda) = -1$. After getting the Lasso path $\hat{\mathbf{B}}_{*,j}(\lambda)$, the estimator (13) can be computed iteratively by

$$\hat{\sigma}_j^2 \leftarrow \hat{\mathbf{B}}_{*,j}^T \hat{\Sigma}_{*,j} \hat{\mathbf{B}}_{*,j}, \quad \lambda \leftarrow \hat{\sigma}_j \lambda_0, \quad \hat{\mathbf{B}}_{*,j} \leftarrow \hat{\mathbf{B}}_{*,j}(\lambda). \quad (17)$$

It is apparent from the above steps that the penalty hyperparameter $\lambda$ is updated in the iterations.

*Hyper-parameter Selection.* We provide two choices of the initial penalty hyperparameter $\lambda_0$:

- Satisfy union bound (Theorem A.2) when:
$$\lambda_0 = \tau\sqrt{4T^{-1}\log p} \text{ for } \tau > 1. \quad (18)$$

- Satisfy probabilistic error bound (Theorem A.3) when:
$$\lambda_0 = \tau L_T(k/p) \text{ for } 1 < \tau \leq \sqrt{2}, \quad (19)$$

where $k$ is a real solution of $k = L_1^4(k/p) + 2L_1^2(k/p)$, $L_a(s) = a^{-1/2}\Phi^{-1}(1-s)$, and $\Phi^{-1}(s)$ is the standard normal quantile function.

## 4 Theoretical Analysis

In this section, we study the theoretical properties of the proposed estimator. Our theoretical analysis can be divided into three parts. Firstly, we present several theorems to validate the selection of the initial penalty hyperparameter $\lambda_0$. Secondly, we provide the selection criteria for the kernel function and discuss the estimation bias after weighting the sample matrix $\mathbf{X}$. Due to the space limit, the other relevant theorems, proofs, and mathematical details are moved to the appendix.

### 4.1 Tuning-free Property

We denote the true covariance matrix and precision matrix as $\Sigma^*$ and $\Omega^*$. Note that we omit the superscript to simplify the representation. First, we consider the capped $\ell_1$ sparsity and the invertibility conditions as follows:

(i) Capped $\ell_1$ sparsity condition: For a certain $\epsilon_0$, $\lambda_0^*$ not depending on $j$ and an index set $\mathcal{P}_j \subset \{1, 2, \ldots, p\}\backslash\{j\}$, the capped $\ell_1$ sparsity of the $j$ th column is defined as

$$|\mathcal{P}_j| + \sum_{k \neq j, k \notin \mathcal{P}_j} \frac{\left|\Omega_{kj}^*\right|}{\left(\Omega_{jj}^*\right)^{1/2}\lambda_0^*} \leq a_j.$$

In the $\ell_0$ sparsity case where $\mathcal{P}_j = \{k : k \neq j, \Omega_{kj}^* \neq 0\}$, we may define $a_j = |\mathcal{P}_j| + 1$ as the degree of the $j$-th node in the graph

induced by the matrix $\Omega^*$. In this scenario, the maximum degree $d$ is given by $d = \max_j(1 + |S_j|)$.

(ii) Invertibility condition: Let $\mathbf{S}$ be the diagonal elements of $\Sigma^*$ and $\mathbf{R}^* = \mathbf{S}^{-1/2}\Sigma^*\mathbf{S}^{-1/2}$. Further, let $\mathcal{P}_j \subseteq Q_j \subseteq \{1, 2, \ldots, p\}\backslash\{j\}$. The invertibility condition is defined as

$$\inf_j \left\{ \frac{\mathbf{u}^T\mathbf{R}_{-j,-j}\mathbf{u}}{\left\|\mathbf{u}_{Q_j}\right\|_2^2} : \mathbf{u} \in \mathbb{R}^p, \mathbf{u}_{Q_j} \neq 0, 1 \leq j \leq p \right\} \geq c_*$$

with a fixed constant $c_* > 0$. Note that the invertibility condition holds if the spectral norm of $(\mathbf{R}^*)^{-1} = \mathbf{S}^{1/2}\Omega^*\mathbf{S}^{1/2}$ is bounded (i.e., $\left\|\mathbf{R}^{-1}\right\|_2 \leq c_*^{-1}$).

THEOREM 4.1. *Let $\hat{\Omega}$ be the scaled Lasso estimators defined in (15) below with penalty level $\lambda_0 = A\sqrt{4(\log p)/n}$, $A > 1$, based on $T$ iid observations from $N(0, \Sigma^*)$. Suppose $d^2(\log p)/n \to 0$. Then,*

$$\left\|\hat{\Omega} - \Omega^*\right\|_2 = O_P(1)d\sqrt{(\log p)/n} = o(1), \quad (20)$$

*where $\|\cdot\|_2$ is the spectrum norm (the $\ell_2$ matrix operator norm).*

THEOREM 4.2. *Suppose $\hat{\Sigma}$ is the sample covariance matrix of $n$ iid $N(0, \Sigma^*)$ vectors. Let $\Omega^* = (\Sigma^*)^{-1}$ and $\Omega^*$ be the inverses of the population covariance and correlation matrices. Let $\hat{\Omega}$ be their scaled Lasso estimators defined in (15) with a penalty level $\lambda_0 = A\sqrt{4(\log p)/T}$, $A > 1$. Suppose the capped $\ell_1$ sparsity condition and invertibility condition hold with $\epsilon_0 = 0$ and $\max_{j \leq p}(1 + a_j)\lambda_0 \leq c_0$ for a certain constant $c_0 > 0$ depending on $c_*$ only. Then, the spectrum norm of the errors is bounded by*

$$\left\|\hat{\Omega} - \Omega^*\right\|_2 \leq \left\|\hat{\Omega} - \Omega^*\right\|_1$$
$$\leq C\left(\max_{j \leq p}\left(\left\|S_{-j}^{-1}\right\|_\infty \Omega_{jj}^*\right)^{1/2} a_j\lambda_0 + \left\|\Omega^*\right\|_1 \lambda_0\right), \quad (21)$$

*with large probability, where $C$ is a constant depending on $\{c_0, c_*, A\}$ only. Moreover, the term $\|\Omega^*\|_1 \lambda_0$ can be replaced by*

$$\max_{j \leq p}\left\|\Omega_{*,j}^*\right\|_1 a_j\lambda_0^2 + \tau_T(\Omega^*),$$

*where $\tau_T(M) = \inf\left\{\tau : \sum_j \exp\left(-T\tau^2/\left\|M_{*,j}\right\|_1^2\right) \leq 1/e\right\}$ for a matrix $M$.*

THEOREM 4.3. *Let $k > 0$. Suppose $\varepsilon \sim N(0, \sigma^2 I_T)$. (i) $\lambda_* = \sigma L_T(k/p)$, and*

$$A - 1 > A_1 \geq \left(\frac{4k/m}{L_1^4(k/p) + 2L_1^2(k/p)}\right)^{1/2} + \frac{L_1(\varepsilon/p)}{L_1(k/p)}\left(\frac{\kappa_+(m)}{m}\right)^{1/2}.$$

*with at least probability $1 - \varepsilon/p - 2|B^c|k/p$. (ii) Let $\lambda_0^* = L_{T-3/2}(k/p)$, $\varepsilon_n = e^{1/(4T-6)^2} - 1$, and*

$$A - 1 > A_1 \geq \left(\frac{(1 + \varepsilon_n)4k/m}{L_1^4(k/p) + 2L_1^2(k/p)}\right)^{1/2}$$
$$+ \left(\frac{L_1(\varepsilon/p)}{L_1(k/p)} + \frac{1 + \varepsilon_T}{L_1(k/p)\sqrt{2\pi}}\right)\left(\frac{\kappa_+(m)}{m}\right)^{1/2}. \quad (22)$$

*Then $\hat{\Omega}$ achieves a consistent estimation with at least probability $1 - 2\varepsilon/p - 2|B^c|k/p$ (See Appendix for more details).*

## 4.2 Selection of the Kernel Function

We assume that the kernel function $K(\cdot)$ has compact support $[-1, 1]$. It is known that the precision matrix is the inverse of the covariance matrix. The estimation bias of the covariance estimation, $\hat{\Sigma}^{(t)} - \Sigma^{(t)}$, will directly impact the estimated precision matrix. Here $\hat{\Sigma}^{(t)} = \text{cov}(\mathbf{X}^{(t)})$, $t = 1, 2, \ldots, T$. For the $t$−th time point and $(i, j)$-th entry, we have

$$\|\hat{\Sigma}_{ij}^{(t)} - \Sigma_{ij}^{(t)}\| \leq \|\hat{\Sigma}_{ij}^{(t)} - \mathbb{E}\hat{\Sigma}_{ij}^{(t)}\| + \|\Sigma_{ij}^{(t)} - \mathbb{E}\hat{\Sigma}_{ij}^{(t)}\| \qquad (23)$$

LEMMA 4.4. *Suppose there exists $C > 0$ such that*

$$\max_{i,j} \sup_t \left|\Sigma_{ij}^{(t)}\right| \leq C.$$

*where $\Sigma_{ij}^{(t)}$ is the $(i, j)$-th entry of the true covariance matrix $\Sigma^{(t)}$. Then for $K(\cdot)$ that satisfies*

$$\sup_{t \in \{0,1,\ldots,T\}} K\left(\frac{t}{hT}\right) = O\left(\frac{1}{h^4}\right), \qquad (24)$$

*we have*

$$\sup_{t \in \{0,1,\ldots,T\}} \max_{i,j} \left|\mathbb{E}\hat{\Sigma}_{ij}^{(t)} - \Sigma_{ij}^{(t)}\right| = O(h) + O\left(\frac{1}{T^2 h^5}\right).$$

LEMMA 4.5. *For $\epsilon < C_0$, we have*

$$P(\|\hat{\Sigma}_{ij}^{(t)} - \mathbb{E}\hat{\Sigma}_{ij}^{(t)}\| > \epsilon) \leq \exp\{-C_1 T h \epsilon^2\}.$$

*where $c_0, c_1$ are constants (See more details in Appendix).*

Therefore, we can bound the covariance estimation with Lemma A.4 and Lemma A.5. This conclusion confirms the validity of (8). It is noticed that most smooth kernel functions including the Gaussian kernel satisfy (33).

## 5 Experiment

### 5.1 Experimental Setting

*Implementation Environment.* All experiments are performed on a machine with an Intel Core i9-10910 ten-core 3.6 GHz CPU and 64 GB RAM.

*Metric.* We use the averaged Frobenius norm $\|\hat{\Omega}^{(t')} - \Omega^{*(t')}\|_F$, Spectrum norm $\|\hat{\Omega}^{(t')} - \Omega^{*(t')}\|_2$, and Matrix $\ell_1$ norm $\|\hat{\Omega}^{(t')} - \Omega^{*(t')}\|_1$, where $\Omega^*$ is the true precision matrix and $t' \in \mathcal{T}$. Here $\mathcal{T} \subset \{1, 2, \ldots, T\}$ is a randomly selectd subset and $|\mathcal{T}| = 10$. MCC is widely used in machine learning as a measure of binary classifiers, defined as follows:

$$\text{MCC} = \frac{\text{TP} \times \text{TN} - \text{FP} \times \text{FN}}{\sqrt{(\text{TP} + \text{FP})(\text{TP} + \text{FN})(\text{TN} + \text{FP})(\text{TN} + \text{FN})}}$$

where the true positive (TP), true negative (TN), false positive (FP), and false negative (FN) values indicate the number of true nonzero entries, true-zero entries, false nonzero entries, and false zero entries, respectively. It produces a high score if the classifier generates desirable estimations.

*Baseline.* We compare our method NoTeNet with the following baselines: 1) NoTeNet-unweighted, NoTeNet without the utilization of the weight matrix, to affirm the weight matrix's vital contribution to the estimation; 2) QUIC-Dependency, a SOTA method [13] assuming temporal dependency (Using (8)), which requires manual hyperparameter tuning; 3) QUIC-Independency, a SOTA method

based on temporal independency assumptions that also necessitates hyperparameter tuning.

Furthermore, to assess the effect of the relationship matrix in $S(\theta)$ on NoTeNet, we introduce a new metric $recall = \text{TP}/(\text{TP+FN})$ and implement our method on the synthetic datasets with two different conditions: 1) NoTeNet($recall = 40\%$), our method with only 40% correct connections in the relationship matrix $S(\theta)$; 2) NoTeNet($recall = 80\%$).

*Explanations about baseline choice.* We choose QUIC as the main baseline since it continues to be widely utilized in current research and serves as a critical benchmark for new methodologies within the realm of state-of-the-art works. [18] utilizes the QUIC method to detect structural changes in high-dimensional Gaussian graphical models. QUIC's ability to perform fast and accurate estimation underpins the methodology for identifying change-points in the graphical model structure over time. Similarly, studies like those in recent works like [20, 28] also employ QUIC for various analytical tasks downstream. Additionally, both [45] and [30] utilize QUIC as a main baseline for comparison.

*Relationship Matrix Simulation.* Specifically, the relationship matrices are created as follows. $S(\theta^{(t)})$ are constructed by setting its entries to one where $\Omega^{*(t)}$ has zero entries and drawing from a uniform distribution $\mathcal{U}(0, 1)$ for a proportion of nonzero entries of $\Omega^{*(t)}$. The proportion depends on the metric $recall$. For the rest nonzero entries, we also set the corresponding entries of $S(\theta^{(t)})$ to one.

*Simulated Datasets.* We illustrate the efficiency of our approach through a simulated scenario. This graph changes over time, guided by the Erdős-Rényi random graph model principles. We start with $\Omega = 0.25\mathbf{I}_{p \times p}$, where $p = 50, 100, 200, 300, 400$. Subsequently, we choose $p/10$ edges at random and adjust $\Omega$ in the following manner: for each newly added edge $(i, j)$, we select a positive weight $w$ uniformly from the range $[0.1, 0.3]$. We then decrease $\Omega_{ij}$ and $\Omega_{ji}$ by $w$, while $\Omega_{ii}$ and $\Omega_{jj}$ are increased by the same amount, ensuring that $\Sigma$ remains positive definite.

As the simulation progresses, when an edge is removed, we implement the reverse of the initial procedure based on the edge's weight. The first 50 edges are allocated weights, after which we systematically alter the graph's structure in a cyclic manner: Every 100 discrete time interval, we eliminate five edges and introduce five new ones. For every one of these new edges, a specific target weight is determined. Over the next 100 time intervals, the weight of each new edge is adjusted incrementally to achieve a smooth transition. In a similar vein, the weight of each edge set to be removed gradually reduces to zero over the same period. As a result, the graph consistently maintains around $1.1p$ edges, with $0.1p$ of those edges undergoing smooth weight adjustments.

## 5.2 Performance Evaluation

In this section, we mainly evaluate the accuracy performance of our method and compare its performance with the other baselines. We fix the number of time points $T = 200$ and vary the dimension in $\{50, 100, 200, 300, 400\}$.

As illustrated in Table 1 and Table 2, we compare NoTeNet ($recall = 0.8$) and NoTeNet ($recall = 0.4$) against other baselines

**Table 1: Comparison of estimation error in terms of Frobenius Norm and Spectrum Norm. Bold and underline represent the first and second rankings respectively.**

| | FROBENIUS NORM | | | | | SPECTURM NORM | | | | |
|---|---|---|---|---|---|---|---|---|---|---|
| P | QUIC-I | QUIC-D | NoTeNet-U | NoTeNet (RECALL=0.4) | NoTeNet (RECALL=0.8) | QUIC-I | QUIC-D | NoTeNet-U | NoTeNet (RECALL=0.4) | NoTeNet (RECALL=0.8) |
| 50 | 4.01 | 2.15 | **1.43** | 1.77 | 1.74 | 2.06 | 0.70 | 70.28 | **0.67** | 0.70 |
| 100 | 8.73 | 3.10 | 2.16 | 2.56 | **0.83** | 5.81 | 0.78 | 93.76 | 0.64 | **0.63** |
| 200 | 11.48 | 4.40 | 8.50 | 3.95 | **3.80** | 6.56 | 0.84 | 91.18 | 0.78 | **0.77** |
| 300 | 15.59 | 5.16 | 32.52 | 4.52 | **3.23** | 10.84 | **0.73** | 56.28 | 0.85 | 0.82 |
| 400 | 14.20 | 5.91 | 96.48 | 5.99 | **4.56** | 8.98 | **0.79** | 120.24 | 1.02 | 0.94 |

**Table 2: Comparison of estimation error in terms of $L_1$ Norm and MCC.Bold and underline represent the first and second rankings respectively.**

| | $L_1$ NORM | | | | | MCC | | | | |
|---|---|---|---|---|---|---|---|---|---|---|
| P | QUIC-I | QUIC-D | NoTeNet-U | NoTeNet (RECALL=0.4) | NoTeNet (RECALL=0.8) | QUIC-I | QUIC-D | NoTeNet-U | NoTeNet (RECALL=0.4) | NoTeNet (RECALL=0.8) |
| 50 | 2.32 | 0.93 | 70.91 | 1.21 | **0.82** | 0.0418 | 0.36 | 0.69 | 0.55 | **0.70** |
| 100 | 10.94 | 1.73 | 114.88 | **1.05** | 1.48 | 0.0243 | 0.18 | 0.39 | 0.45 | **0.63** |
| 200 | 10.70 | 1.79 | 102.35 | 2.11 | **1.60** | 0.0067 | 0.17 | 0.19 | 0.42 | **0.46** |
| 300 | 20.19 | 1.35 | 88.51 | 2.81 | **1.15** | 0.0082 | 0.17 | 0.12 | 0.35 | **0.35** |
| 400 | 26.18 | **1.92** | 122.30 | 3.85 | 2.00 | 0.0057 | 0.15 | 0.10 | 0.25 | **0.28** |

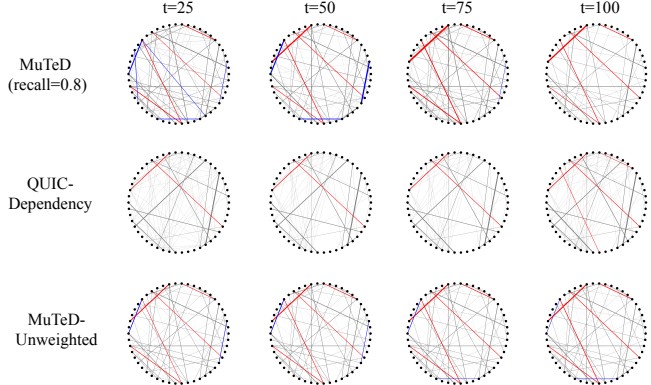

**Figure 2: Visualization of the dynamic networks predicted by NoTeNet ($recall = 0.8$), NoTeNet-Unweighted and QUIC-Dependency. To highlight the prediction of the time-varying edges, we artificially amplify the weights of the added edges (red) and deleted edges (blue).**

across three metrics. The penalty parameters for QUIC-Independency (QUIC-I) and QUIC-Dependency (QUIC-D) are manually tuned to optimize performance. Our method outperforms all baselines in most cases. A comparison between QUIC-I and QUIC-D reveals that QUIC-D, which fully leverages data from adjacent timestamps, demonstrates superior performance, thus validating our time-varying weighting techniques (8). When comparing NoTeNet-Unweighted (NoTeNet-U) with QUIC-D, both of which utilize a time-varying weighted matrix, QUIC-D excels in norm metrics, while

NoTeNet-U shows better performance in the MCC value. This indicates that NoTeNet-U is more effective in distinguishing between zero and non-zero entries, although it does not predict precise edge values as well. Between NoTeNet-U and our NoTeNet($recall = 0.8$), both of which do not require tuning, our method exhibits superior performance across all metrics, thanks to the use of the relationship matrix derived from EEG time series data. To evaluate the impact of the relationship matrix, we vary the value of $recall$. NoTeNet($recall = 0.4$) performs worse than NoTeNet($recall = 0.8$) at most time, but still performs better than the other baselines in MCC value.

Figure 2 shows that our method MuTeD performs better than the other baselines. QUIC-Dependency is unable to capture all new edges with increasing weights and all deleted edges with decreasing weights. In the case of MuTeD-Unweighted, it is able to capture all new edges but fails to capture vanishing edges. As the values of the deleted edges decrease over time, our method detects fewer edges, which is consistent with our expectation.

## 5.3 Application to Simultaneous Medical Sensor dataset

*Dataset Description.* As mentioned above, whole-brain functional connectomes offer significant potential for understanding human brain activity across a range of cognitive, developmental, and pathological states. Resting-state (rs) functional Magnetic Resonance Imaging (fMRI) studies have led to the brain being considered at a macroscopic scale as a set of interacting regions. Due to the low temporal and spatial resolution of fMRI data, it is common to adopt a multimodal approach that integrates fMRI data with other modalities.

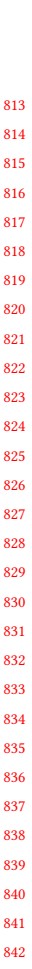

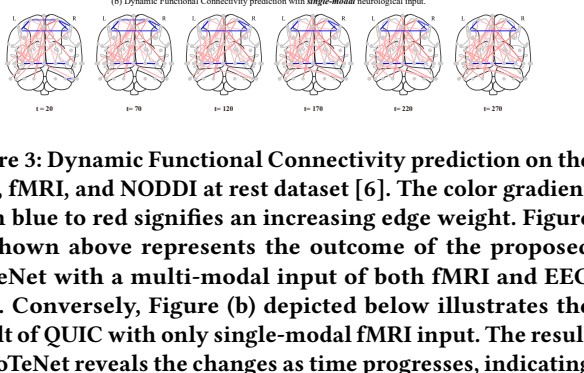

(a) Dynamic Functional Connectivity prediction with *multi-modal* neurological input.

$t = 20 \quad t = 70 \quad t = 120 \quad t = 170 \quad t = 220 \quad t = 270$

(b) Dynamic Functional Connectivity prediction with *single-modal* neurological input.

$t = 20 \quad t = 70 \quad t = 120 \quad t = 170 \quad t = 220 \quad t = 270$

**Figure 3: Dynamic Functional Connectivity prediction on the EEG, fMRI, and NODDI at rest dataset [6]. The color gradient from blue to red signifies an increasing edge weight. Figure (a) shown above represents the outcome of the proposed NoTeNet with a multi-modal input of both fMRI and EEG data. Conversely, Figure (b) depicted below illustrates the result of QUIC with only single-modal fMRI input. The result of NoTeNet reveals the changes as time progresses, indicating a successful capturing of the dynamic state of the neural connectivity. In comparison, the prediction of QUIC remains basically unchanged, revealing poor temporal dependency.**

To evaluate the performance of our method in such real-world neuroscience research involving multi-modal data, we utilize the EEG, fMRI, and NODDI at rest dataset [6] developed by F. Deligianni et.al., which is a comprehensive collection of neuroimaging data that encompasses EEG, fMRI, and NODDI (neurite orientation dispersion and density imaging) measurements.

We select the fMRI and EEG modalities for the experiment. The fMRI data for each subject contains 300 volumes, $TR/TE = 2160/30$ ms, with voxel size being $3.3 \times 3.3 \times 4.0$ mm. EEG data is recorded with a 64-channel MR-compatible electrode cap at a native frequency of 1000 Hz. We adhere to the preprocessing procedures suggested in [6], which contain common fMRI motion correction and EEG artifact removal using the FSL [36] and EEGLAB [7] kit respectively. ROIs are delineated as the cerebral cortex areas corresponding to the electrodes of EEG (excluding the ECG channel). In short, the preprocessed and aligned fMRI and EEG signals exhibit a shape of (300, 63) and (300, 540, 63) respectively.

*Result Visualization.* In this study, we performed a Dynamic Functional Connectivity prediction on the EEG, fMRI, and NODDI at rest dataset. Figure 2 visualized the connectivity of the ROIs, i.e. the connectivity of the cerebral cortex beneath where the EEG electrodes are located. We filter out the weak connections to demonstrate the main predicted functional connectivity in each time step. The edges between ROIs are visualized with a color gradient ranging from blue to red, with the intensity of the color signifying the increasing edge weight.

Subfigure (a) in Figure 2 represents the outcome of our proposed method, denoted as NoTeNet, which utilizes a multi-modal input of both fMRI and EEG data. The visualization of results from NoTeNet reveals an interesting pattern of changes as time progresses. The color gradient shifts, indicating a dynamic alteration in the edge weights. This successful capturing of the dynamic state of neural connectivity suggests that NoTeNet is capable of tracking the

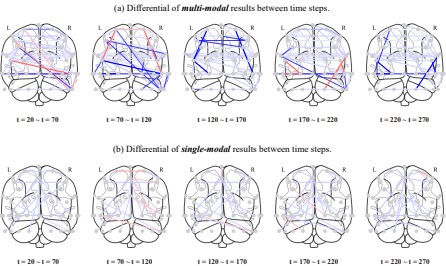

(a) Differential of *multi-modal* results between time steps.

$t = 20 \sim t = 70 \quad t = 70 \sim t = 120 \quad t = 120 \sim t = 170 \quad t = 170 \sim t = 220 \quad t = 220 \sim t = 270$

(b) Differential of *single-modal* results between time steps.

$t = 20 \sim t = 70 \quad t = 70 \sim t = 120 \quad t = 120 \sim t = 170 \quad t = 170 \sim t = 220 \quad t = 220 \sim t = 270$

**Figure 4: The corresponding differential of multi-modal and single-modal results. The results of NoTeNet in (a) demonstrate more significant temporal variability when compared to that of QUIC in (b). This underscores the higher capability of NoTeNet in the temporal dependency prediction of dynamic functional connectivity as compared to the single-modal method.**

temporal evolution of brain connectivity, providing a more comprehensive and nuanced understanding of brain function.

On the contrary, subfigure (b) in Figure 2 illustrates the result of the QUIC method, which employs only single-modal fMRI input. In stark contrast to the dynamic changes observed with NoTeNet, the prediction outcomes of QUIC remain essentially unchanged over time. The absence of significant color gradient shifts in the QUIC results reveals a poor temporal dependency. This suggests that the QUIC method may not be as effective in capturing the dynamic changes in neural connectivity over time.

As presented in Figure 4, the corresponding differential of multi-modal results have significantly higher absolute values, which further emphasizes the higher capability of NoTeNet in temporal dependency prediction of dynamic functional connectivity than the single-modal method.

Overall, the comparison of these two methods highlights the potential advantages of our proposed NoTeNet method in capturing the dynamic state of neural connectivity, underscoring the importance of incorporating multi-modal data inputs and the ability to track changes over time when studying brain connectivity.

## 6 Conclusion

In this paper, we introduce NoTeNet, a tuning-free dynamic dependence network inference method. the challenges of temporal independency assumptions, manual hyperparameter tuning, and the underutilization of multimodal data in dynamic network prediction. By leveraging mutual information and a kernel-based weighting strategy, NoTeNet effectively integrates data across modalities, significantly enhancing prediction accuracy and reducing manual intervention. Experiments on synthetic and real-world datasets, including EEG-fMRI data, demonstrated NoTeNet's superior performance in capturing time-varying dependencies compared to existing methods. Our framework's generality and efficiency make it suitable for a wide range of web data applications, such as neuroscience analysis, offering a promising solution for dynamic network analysis. Future work could explore extending the framework to other domains and investigating potential improvements in computational efficiency.

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

## A  Theoretical Analysis

In this section, we study the theoretical properties of the proposed estimator. Our theoretical analysis can be divided into three parts. Firstly, we present several theorems to validate the selection of the initial penalty hyperparameter $\lambda_0$. Secondly, we provide the selection criteria for the kernel function and discuss the estimation bias after weighting the sample matrix $\mathbf{X}$.

### A.1  Tuning-free Property

We denote the true covariance matrix and precision matrix as $\Sigma^*$ and $\Omega^*$. Note that we omit the superscript to simplify the representation. First, we consider the capped $\ell_1$ sparsity and the invertibility conditions as follows:

(i) Capped $\ell_1$ sparsity condition: For a certain $\epsilon_0, \lambda_0^*$ not depending on $j$ and an index set $\mathcal{P}_j \subset \{1, 2, \ldots, p\} \setminus \{j\}$, the capped $\ell_1$ sparsity of the $j$ th column is defined as

$$\left|\mathcal{P}_j\right| + \sum_{k \neq j, k \notin \mathcal{P}_j} \frac{\left|\Omega_{kj}^*\right|}{\left(\Omega_{jj}^*\right)^{1/2} \lambda_0^*} \leq a_j.$$

In the $\ell_0$ sparsity case where $\mathcal{P}_j = \{k : k \neq j, \Omega_{kj}^* \neq 0\}$, we may define $a_j = |\mathcal{P}_j| + 1$ as the degree of the $j$-th node in the graph induced by the matrix $\Omega^*$. In this scenario, the maximum degree $d$ is given by $d = \max_j(1 + |S_j|)$.

(ii) Invertibility condition: Let $\mathbf{S}$ be the diagonal elements of $\Sigma^*$ and $\mathbf{R}^* = \mathbf{S}^{-1/2} \Sigma^* \mathbf{S}^{-1/2}$. Further, let $\mathcal{P}_j \subseteq Q_j \subseteq \{1, 2, \ldots, p\} \setminus \{j\}$. The invertibility condition is defined as

$$\inf_j \left\{ \frac{\mathbf{u}^T \mathbf{R}_{-j,-j} \mathbf{u}}{\left\|\mathbf{u}_{Q_j}\right\|_2^2} : \mathbf{u} \in \mathbb{R}^p, \mathbf{u}_{Q_j} \neq 0, 1 \leq j \leq p \right\} \geq c_*$$

with a fixed constant $c_* > 0$. Note that the invertibility condition holds if the spectral norm of $(\mathbf{R}^*)^{-1} = \mathbf{S}^{1/2} \Omega^* \mathbf{S}^{1/2}$ is bounded (i.e., $\left\|\mathbf{R}^{-1}\right\|_2 \leq c_*^{-1}$).

**Theorem A.1.** *Let $\hat{\Omega}$ be the scaled Lasso estimators defined in (11) below with penalty level $\lambda_0 = A\sqrt{4(\log p)/T}, A > 1$, based on $T$ iid observations from $N(0, \Sigma^*)$. Suppose $d^2(\log p)/T \to 0$. Then,*

$$\left\|\hat{\Omega} - \Omega^*\right\|_2 = O_P(1)d\sqrt{(\log p)/T} = o(1). \tag{25}$$

*where $\|\cdot\|_2$ is the spectrum norm (the $\ell_2$ matrix operator norm).*

Theorem 1 establishes that for the convergence of $\widehat{\Omega}$ in the spectral norm, there is a straightforward boundedness requirement on the spectral norm of $\Omega^*$. This condition is satisfied when the sample size $T$ significantly exceeds $d^2 \log p$.

**Theorem A.2.** *Suppose $\hat{\Sigma}$ is the sample covariance matrix of $n$ iid $N(0, \Sigma^*)$ vectors. Let $\Omega^* = (\Sigma^*)^{-1}$ and $\Omega^*$ be the inverses of the population covariance and correlation matrices. Let $\hat{\Omega}$ be their scaled Lasso estimators defined in (12) with a penalty level $\lambda_0 = A\sqrt{4(\log p)/T}, A > 1$. Suppose the capped $\ell_1$ sparsity condition and invertibility condition hold with $\epsilon_0 = 0$ and $\max_{j \leq p}(1 + a_j)\lambda_0 \leq c_0$ for a certain constant $c_0 > 0$ depending on $c_*$ only. Then, the spectrum norm of the errors is bounded by*

$$\left\|\hat{\Omega} - \Omega^*\right\|_2 \leq \left\|\hat{\Omega} - \Omega^*\right\|_1$$
$$\leq C\left(\max_{j \leq p}\left(\left\|S_{-j}^{-1}\right\|_\infty \Omega_{jj}^*\right)^{1/2} a_j \lambda_0 + \left\|\Omega^*\right\|_1 \lambda_0\right), \tag{26}$$

*with large probability, where $C$ is a constant depending on $\{c_0, c_*, A\}$ only. Moreover, the term $\|\Omega^*\|_1 \lambda_0$ can be replaced by*

$$\max_{j \leq p}\left\|\Omega_{*,j}^*\right\|_1 a_j \lambda_0^2 + \tau_T(\Omega^*), \tag{27}$$

*where $\tau_T(M) = \inf\left\{\tau : \sum_j \exp\left(-T\tau^2 / \|M_{*,j}\|_1^2\right) \leq 1/e\right\}$ for a matrix $M$.*

**Proposition 1.** *Consider $\Omega^*$, a nonnegative definite matrix, and define $\Sigma^* = (\Omega^*)^{-1}$ and $\beta = -\Omega^*(\mathrm{diag}\,\Omega^*)^{-1}$. Let $\widehat{\Omega}$ be as defined in equations (12), derived from certain $\widehat{\beta}$ and $\widehat{\sigma}_j$ that meet the criteria*

$$\left|\frac{\sigma_j^*}{\widehat{\sigma}_j} - 1\right| \leq C_1 a_j \lambda_0^2, \quad \sum_{k \neq j} \hat{\Sigma}_{kk}^{1/2}\left|\widehat{\beta}_{k,j} - \beta_{k,j}\right|\sqrt{\Omega_{jj}^*} \leq C_2 a_j \lambda_0. \tag{28}$$

*Assume that the conditions*

$$\left|\Omega_{jj}^*\left(\sigma_j^*\right)^2 - 1\right| \leq C_0 \lambda_0, \quad \max_j\left|\left(\hat{\Sigma}_{jj}/\Sigma_{jj}^*\right)^{-1/2} - 1\right| \leq C_0 \lambda_0 \tag{29}$$

*are satisfied, and that $\max 4C_0\lambda_0, 4\lambda_0, C_1 s_j \lambda_0 \leq 1$. Under these assumptions, equations (26) are valid with a constant $C$ that depends solely on $C_0, C_2$. Furthermore, if $T\Omega_{jj}^*(\sigma_j^*)^2 \sim \chi_T^2$, then the term $\lambda_0 |\Omega^*|_1$ in (26)) can be substituted by equation (27) with high probability.*

*Proof for Theorem A.2.* To utilize Proposition 1, it's crucial to confirm the validity of conditions (28) and (29). Given that $\Omega_{jj}^*\left(\sigma_j^*\right)^2$ and $\hat{\Sigma}_{jj}/\Sigma_{jj}^*$ both approximate $\chi_T^2/T$, condition (29) holds when $\lambda_0$ is proportional to $\sqrt{(\log p)/T}$. Additionally, the probability $P\{(1 - \epsilon_0)^2 \leq \chi_T^2/T \leq (1 + \epsilon_0)^2\}$ being less than $\varepsilon/p$ is feasible with sufficiently small values of $\epsilon_0$ and $\varepsilon$, considering the assumption that $\sqrt{(\log p)/T} = \lambda_0/(2A)$ is notably small. We choose $\epsilon_0 = 0$ in the capped $\ell_1$ sparsity since it does not affect the scaling of $a_j$.

Considering $\hat{\Sigma}_{kk}^{1/2}\beta_k$ as the regression coefficient in (11) for the normalized design vector $\hat{\Sigma}_{kk}^{-1/2}x_k$ (for $k \neq j$), Theorem 8 in [38] applies with a probability of $1 - 3\varepsilon/p$ for each $j$, that provides bounded ratios of estimated to true noise levels and explicit upper bounds on prediction and estimation errors under specific conditions. These probabilities are determined assuming $\lambda_0 = A\sqrt{4(\log p)/T}, A_1 = 0$ and $\varepsilon \approx 1/\sqrt{\log p}$. Using the union bound, the results of Theorem 8 are collectively valid for all $j$ with a probability of $1 - 3\varepsilon$. Condition (28) is included in Theorem 8's results, asserting that $M_\sigma^*$ and $M_1^*$ remain uniformly bounded across the $p$ regression scenarios with high probability.

The uniform boundedness of $M_\sigma^*$ and $M_1^*$ is verified when $A_1 = 0, B_j = S_j, m_j = 0$ are set, and the matrices $\{\hat{\Sigma}, \Sigma^*\}$ are substituted by $\left\{\hat{R}_{-j,-j}, R_{-j,-j}^*\right\}$. The Gram matrix for the regression setup in (11) is the random and $j$-dependent $\hat{R}_{-j,-j}$. Then we have

$$\max_{k \neq j}\left\|\hat{R}_{k,-j} - R_{k,-j}^*\right\|_\infty \leq \max_{j,k}\left|\hat{R}_{k,j} - R_{k,j}^*\right| \leq L_T\left(5\varepsilon/p^2\right) \tag{30}$$

with a likelihood of $1 - \varepsilon$. Setting $L_T\left(5\varepsilon/p^2\right) = 2\sqrt{(\log p)/T}$, with $\varepsilon \approx 1/\sqrt{\log p}$. The second stipulation that $c_* |u_S|\, 2^2 \leq u^T R_{-j,-j}^* u$ follows from the invertibility condition, and the third condition mandates that $\max j \leq p\lambda_0 s*, j \leq c_0$.

THEOREM A.3. *Let $k > 0$. Suppose $\varepsilon \sim N\left(0, \sigma^2 I_T\right)$. (i) $\lambda_* = \sigma L_T(k/p)$, and*

$$A - 1 > A_1 \geq \left(\frac{4k/m}{L_1^4(k/p) + 2L_1^2(k/p)}\right)^{1/2} + \frac{L_1(\varepsilon/p)}{L_1(k/p)}\left(\frac{\kappa_+(m)}{m}\right)^{1/2}.$$

*with at least probability $1-\varepsilon/p-2\,|B^c|\,k/p$. (ii) Let $\lambda_0^* = L_{T-3/2}(k/p)$, $\varepsilon_n = e^{1/(4T-6)^2} - 1$, and*

$$
\begin{aligned}
A - 1 > A_1 \geq &\left(\frac{(1+\varepsilon_n)\,4k/m}{L_1^4(k/p) + 2L_1^2(k/p)}\right)^{1/2} \\
&+ \left(\frac{L_1(\varepsilon/p)}{L_1(k/p)} + \frac{1+\varepsilon_T}{L_1(k/p)\sqrt{2\pi}}\right)\left(\frac{\kappa_+(m)}{m}\right)^{1/2}.
\end{aligned}
\tag{31}
$$

*Then $\hat{\Omega}$ achieves a consistent estimation with at least probability $1 - 2\varepsilon/p - 2\,|B^c|\,k/p$.*

The proof of Theorem A.1 and Theorem A.3 is similar to [38], thus we will not elaborate further.

## A.2 Selection of the Kernel Function

We assume that the kernel function $K(\cdot)$ has compact support $[-1, 1]$. It is known that the precision matrix is the inverse of the covariance matrix. The estimation bias of the covariance estimation, $\hat{\Sigma}^{(t)} - \Sigma^{(t)}$, will directly impact the estimated precision matrix. Here $\hat{\Sigma}^{(t)} = \text{cov}(\mathbf{X}^{(t)})$, $t = 1, 2, \ldots, T$. For the $t$−th time point and $(i, j)$-th entry, we have

$$\|\hat{\Sigma}_{ij}^{(t)} - \Sigma_{ij}^{(t)}\| \leq \|\hat{\Sigma}_{ij}^{(t)} - \mathbb{E}\hat{\Sigma}_{ij}^{(t)}\| + \|\Sigma_{ij}^{(t)} - \mathbb{E}\hat{\Sigma}_{ij}^{(t)}\| \tag{32}$$

LEMMA A.4. *Suppose there exists $C > 0$ such that*

$$\max_{i,j} \sup_t \left|\Sigma_{ij}^{(t)}\right| \leq C.$$

*where $\Sigma_{ij}^{(t)}$ is the $(i, j)$-th entry of the true covariance matrix $\Sigma^{(t)}$. Then for $K(\cdot)$ that satisfies*

$$\sup_{t \in \{0,1,\ldots,T\}} K\left(\frac{t}{hT}\right) = O\left(\frac{1}{h^4}\right), \tag{33}$$

*we have*

$$\sup_{t \in \{0,1,\ldots,T\}} \max_{i,j} \left|\mathbb{E}\hat{\Sigma}_{ij}^{(t)} - \Sigma_{ij}^{(t)}\right| = O(h) + O\left(\frac{1}{T^2 h^5}\right).$$

LEMMA A.5. *For $\epsilon < C_0$, we have*

$$P(\|\hat{\Sigma}_{ij}^{(t)} - \mathbb{E}\hat{\Sigma}_{ij}^{(t)}\| > \epsilon) \leq \exp\{-C_1 T h \epsilon^2\}.$$

*where $C_1 > 0$ and $C_0$ is a constant such that*

$$C_0 = \frac{C_1\left((\Sigma_i^{(t)})^2 (\Sigma_j^{(t)})^2 + (\Sigma_{ij}^{(t)})^2\right)}{\max_{k=1,\ldots,T}\left(2K\left(\frac{k-t}{hT}\right)\Sigma_i^{(k)}\Sigma_j^{(k)}\right)}$$

Therefore, we can bound the covariance estimation with Lemma A.4 and Lemma A.5. This conclusion confirms the validity of (6). It is noticed that most smooth kernel functions including the Gaussian kernel satisfy (33).

*Proof for Theorem A.4.* Without loss of generality, assume that $t = T$. To estimate the sum, we employ the Riemann integral approximation.

$$
\begin{aligned}
\mathbb{E}\hat{\Sigma}_{ij}^{(t)} &= \frac{1}{T}\sum_{k=1}^{T}\frac{2}{h}K\left(\frac{k-t}{hT}\right)\Sigma_{ij}^{(k)} \\
&= \int_k^t \frac{2}{h}K\left(\frac{u-t}{hT}\right)\Sigma_{ij}^{(u)}du + O\left(\frac{2}{h}\sup_{u\in[k,T]}\frac{\left(K\left(\frac{u-t}{hT}\right)\Sigma_{ij}^{(u)}\right)}{T^2}\right) \\
&= 2\int_{-1/h}^0 K(v)\Sigma_{ij}^{(t+hv)}dv + O\left(\frac{1}{T^2 h^5}\right).
\end{aligned}
$$

We now use Taylor's formula to replace $\Sigma_{ij}^{(t+hv)}$ and obtain

$$
\begin{aligned}
&2\int_{-1/h}^0 K(v)\Sigma_{ij}^{(t+hv)}dv \\
&= 2\int_{-1}^0 K(v)\left(\Sigma_{ij}^{(t)} + hv\Sigma_{ij}^{(t)} + \frac{\Sigma_{ij}^{(y(v))}(hv)^2}{2}\right)dv \\
&= \Sigma_{ij}^{(t)} + 2\int_{-1}^0 K(v)\left(hv\Sigma_{ij}^{(t)} + \frac{C(hv)^2}{2}\right)dv
\end{aligned}
$$

where

$$
\begin{aligned}
&2\int_{-1}^0 K(v)\left(hv\Sigma_{ij}^{(t)} + \frac{C(hv)^2}{2}\right)dv \\
&= 2h\Sigma_{ij}^{(t)}\int_{-1}^0 vK(v)dv + \frac{Ch^2}{2}\int_{-1}^0 v^2 K(v)dv \\
&\leq h\Sigma_{ij}^{(t)} + \frac{Ch^2}{4}
\end{aligned}
$$

with $y(v) - t < hv$. Then $\mathbb{E}\hat{\Sigma}_{ij}^{(t)} - \Sigma_{ij}^{(t)} = O(h) + O\left(\frac{1}{T^2 h^5}\right)$ and the lemma holds.

*Proof for Theorem A.5.* Let us define $A_t = \mathbf{X}_{ti}\mathbf{X}_{tj} - \Sigma_{ij}^{(t)}$.

$$
\begin{aligned}
&\mathbf{P}\left(\left|\hat{\Sigma}_{ij}^{(t)} - \mathbb{E}\hat{\Sigma}_{ij}^{(t)}\right| > \epsilon\right) \\
&= \mathbf{P}\left(\sum_{k=1}^{T}\ell_k(t)\mathbf{X}_{ki}\mathbf{X}_{kj} - \sum_{k=1}^{T}\ell_k(t)\Sigma_{ij}^{(k)} > \epsilon\right).
\end{aligned}
$$

where

$$\ell_k(t) = \frac{2}{Th}K\left(\frac{k-t}{hT}\right) \approx \frac{K\left(\frac{k-t}{hT}\right)}{\sum_{i=1}^{T}K\left(\frac{k-t}{hT}\right)}$$

For every $t > 0$, we have by Markov's inequality

$$
\begin{aligned}
\mathbf{P}\left(\sum_{k=1}^{T}T\ell_k(t)A_k > T\epsilon\right) &= \mathbf{P}\left(\exp\left(t\sum_{k=1}^{T}\frac{2}{h}K\left(\frac{k-t}{hT}\right)A_k\right) > e^{Tt\epsilon}\right) \\
&\leq \frac{\mathbb{E}\exp\left(t\sum_{k=1}^{n}\frac{2}{h}K\left(\frac{k-t}{hT}\right)A_k\right)}{e^{Tt\epsilon}}.
\end{aligned}
$$

The lemma holds.

### A.3 Explanation for $S(\cdot)$

In the main text, we let $S(z) = 1 - z$. Normalized mutual information is a quantity that measures a relationship between two random variables that are sampled simultaneously. Higher normalized mutual information indicates a greater degree of dependence between the variables, implying a stronger relationship and better predictability of one variable based on the other. Therefore, the position $(i, j)$ is more likely to have an edge if $\theta_{ij}$ is higher. As we know, the larger the penalty parameter, the stronger the penalty applied, compressing the coefficient towards zero. Thus we inverse the $\theta_{ij}$ with the operator $S(\cdot)$, where $\theta_{ij}$ ranges in $[0, 1]$.

## B More Information about Real-world Dataset

### B.1 Multimedia in Neuroscience

Functional Magnetic Resonance Imaging (fMRI) and Electroencephalography (EEG) are two prominent neuroimaging techniques used to explore and understand brain activity.

fMRI is a technique that measures brain activity by detecting changes in blood flow. When an area of the brain is more active, it consumes more oxygen, and to meet this increased demand, blood flow to that region also increases. This phenomenon is known as the Blood Oxygen Level Dependent (BOLD) contrast. It provides high spatial resolution, offering detailed images of brain structures. It can pinpoint the location of brain activity within millimeters.

EEG, on the other hand, directly measures electrical activity in the brain using dozens of electrode channels placed on the scalp. When neurons fire, they produce electrical signals that can be detected and recorded by EEG. EEG has excellent temporal resolution, on the order of milliseconds. This allows researchers to track changes in brain activity in real time, providing insights into the dynamics of cognitive processes.

Both fMRI and EEG offer valuable insights into brain function, with complementary strengths and weaknesses. Researchers are exploring to obtain a more comprehensive understanding of brain activity using them.

### B.2 Functional Connectivity Network

Functional connectivity (FC) refers to the statistical dependencies or correlations between different brain regions based on their neural activity. It provides insights into how different brain areas communicate and work together during various cognitive tasks or even at rest. Its prediction is often derived from data sources such as functional magnetic resonance imaging (fMRI), electroencephalography (EEG), or other neuroimaging modalities. FC prediction plays a crucial role in neuroscience, clinical diagnosis, and personalized medicine [8, 14, 39].

