# OpenReview forum: "NoTeNet: Normalized Mutual Information-Driven Tuning-free Dynamic Dependence Network Inference Method for Multimodal Data"
_ACM.org/TheWebConf/2025/Conference — WWW 2025 Poster_

### Official Review · Reviewer_MyH5 · 2024-11-03

**Novelty:** 4
**Technical Quality:** 5

**Review:**

This paper is not well aligned to my area of expertise, so any evaluation comes with that heavy caveat.

This paper introduce a Dynamic dependence network inference method for multimodal data, named Normalized mutual information-driven Tuning-free DDN. It mainly address the question regarding computational efficiency, prediction stability and robustness, and modality quality disparity.

Pros:
1. Extensive experiments design to support their statement in the method section.
1. Every extensive and sound theoretical analysis.

Cons:
1. Better to provide a reproducing code.
2. The math formula can be aligned and format better

**Questions:**

This paper is not well aligned to my area of expertise - I would rather leave questions to the other reviewers.

**Reviewer Confidence:**

1: The reviewer's evaluation is an educated guess

**Scope:**

3: The work is somewhat relevant to the Web and to the track, and is of narrow interest to a sub-community

---

### Official Review · Reviewer_oMvU · 2024-11-28

**Novelty:** 5
**Technical Quality:** 5

**Review:**

The work presents a method called NoTeNet, which addresses significant challenges in Dynamic Dependence Network (DDN) inference for multimodal data. The method is grounded in robust theoretical foundations, utilizing normalized mutual information and a tuning-free approach to enhance prediction accuracy and efficiency. The experimental results demonstrate superior performance compared to existing methods, indicating a high-quality contribution to the field.

Pros:
1. The problem is interesting and practical.
2. The paper provides theoretical guarantees for the method's performance, enhancing its credibility.

Cons:
1. The process description is not clear.
2. The paper could benefit from a more in-depth discussion of potential limitations or scenarios where the method may not perform as well.
3. The effectiveness of the method relies on the quality of the auxiliary data used for relationship matrix estimation, which may vary in real-world applications.

**Questions:**

1. Can you elaborate on the theoretical guarantees that support the tuning-free nature of NoTeNet? How does this property compare to other methods that still require some form of tuning, particularly in high-dimensional settings?
2. In your experiments, how did you ensure that the integration of different modalities (e.g., EEG and fMRI) was optimal?
3. While NoTeNet outperformed existing methods in your experiments, can you discuss any specific cases or datasets where the performance was not as strong? What insights can be drawn from these cases?
4. How robust is NoTeNet to varying levels of noise in the auxiliary data? Are there specific thresholds or conditions under which the performance might degrade?

**Reviewer Confidence:**

4: The reviewer is certain that the evaluation is correct and very familiar with the relevant literature

**Scope:**

3: The work is somewhat relevant to the Web and to the track, and is of narrow interest to a sub-community

---

### Official Review · Reviewer_RdCH · 2024-11-28

**Novelty:** 5
**Technical Quality:** 4

**Review:**

Existing methods for DDN inference face significant challenges when processing multimodal data, including low computational efficiency, insufficient prediction stability and robustness. To address these issues, the authors propose NoTeNet, a novel framework that leverages normalized mutual information and kernel functions to construct DDN inference. This framework emphasizes reducing the reliance on manual hyperparameter tuning while simultaneously improving prediction accuracy and computational efficiency.

Pros:

(1) The theoretical foundation is solid. The authors provided comprehensive theoretical proofs for the tuning-free approach, enhancing the reliability of the proposed method.

(2) The methodology is well-designed. The use of normalized mutual information and kernel smoothing is interesting, which brings new insights for DNN inference.

(3) The paper is well-designed.

Cons:

(1) Although NoTeNet demonstrates strong performance, it would be beneficial if the authors could provide a discussion on its behavior in edge cases where the method might fail or underperform (e.g., extreme noise levels in auxiliary modalities).

(2) The authors could better emphasize how NoTeNet compares to other state-of-the-art techniques in practical scenarios beyond what is tested.

(3) The theoretical proofs in the paper involve a substantial number of symbols. Presenting these symbols in a table format would significantly enhance readability and facilitate understanding for the readers.

**Questions:**

Please see above Cons.

**Reviewer Confidence:**

3: The reviewer is confident but not certain that the evaluation is correct

**Scope:**

3: The work is somewhat relevant to the Web and to the track, and is of narrow interest to a sub-community

---

### Official Review · Reviewer_wnv1 · 2024-12-02

**Novelty:** 5
**Technical Quality:** 5

**Review:**

The authors consider the problem of integrating data from across different modalities to aid in inferring dynamic dependence networks. A core assumption is that those dynamic dependence networks undergo smooth transitions or, in other words, that the network at the next time point is expected to be similar to the network at the current time point. As highlighted in Fig. 1, the authors integrate data from two modalities where one of them is designated the target modality (here fMRI data) and the other one is designated the auxiliary modality (here EEG data). The auxiliary modality is first discretised and then transformed into a sequence of relationship matrices using normalised mutual information. As an advantage of their approach, the authors mention increased computational efficiency due to the tuning-free nature of their approach. Applied to a synthetic and a real dataset, the authors show that their method outperforms the chosen baselines in most cases.

I find that Figure 1 provides an excellent high-level overview of the manuscript's story in general. The proposed approach to infer dynamic dependence networks seems novel and its benefits are supported by theoretical analyses and mathematical proofs. The considered problem is certainly important, however, it has not been applied to an application relevant to the web and it is unclear what specific web-related application would benefit from the proposed method.


### Strengths
- **S1** The authors identified important challenges relating to the inference of dynamic dependence networks, namely designing computationally efficient methods with prediction stability and robustness while addressing modality quality disparity.
- **S2** Theoretical analyses and proofs support the benefits of the proposed method.
- **S3** Figure 1 provides a clear high-level overview of the proposed method.

### Weaknesses
- **W1** While the present work considers an important research question regarding the integration of data from different modalities, it is unclear how the work relates to the web. The work mentions web data a couple of times, however, it is unclear how the provided examples of "brain functional connectivity in auditory regions" and "EEG and fMRI data gathered from the Internet of Things" relate to the web. Moreover, the required relevance statement on the first page is missing.
- **W2** Only a few baselines were included in the paper, half of which seem more like an ablation study since they are variants of the proposed approach.
- **W3** The authors mention modality quality disparity as a core issue when integrating data from across different modalities, however, it remains unclear whether the proposed approach addresses this issue.

**Questions:**

- **Q1** Can the authors provide an intuitive explanation of how "the transformation of the auxiliary dataset into a relationship matrix using normalised mutual information" is done? Is it simply that the "correlation" between each nodes' feature vectors at two consecutive time points is measured through NMI or is there more to it?
- **Q2** Connected to the previous question, I am wondering how the NMI values are integrated into the inference? Does this happen essentially in Eq. 13? Then, if I am not mistaken, it would seem that the choice of bins for discretisation should indeed have an effect on the final results.
- **Q3** In section 2.2, it is mentioned in l. 269 that the EEG data is discretised to "reduce the effect of noise". How are the bins for discretisation chosen? How sensitive are the final results to the choice of bins?
- **Q4** NMI is claimed to be more robust to noise than entropy because it is normalised to be between 0 and 1 (l.149)? However, I am not sure I understand this claim since NMI is essentially just relative entropy, normalised by entropy. How does NMI reduce noise?
- **Q5** What metric is used in section 5 to measure prediction performance? L. 690 states that accuracy is used, however, earlier on, the definition of Matthews correlation coefficient (MCC) was provided (l.624).
- **Q6** NoTeNet is claimed to be "automatically set to achieve the optimal convergence rate", but how is this done and how has it been confirmed that this is indeed the case?
- **Q7** How are dynamic dependence networks different from temporal graphs, or are they just two different names for the same thing?
- **Q8** What is the MuTeD method that is mentioned in Fig. 2 and l. 791? Is it perhaps the previous name of the proposed method before it was renamed to NoTeNet?
- **Q9** While "web data" was mentioned a couple of times in the manuscript, the approach has not been applied to web data. Could the authors provide a use case in the context of web data where integrating data from across different modalities as described in the manuscript is essential and would lead to novel insights?


### Minor points

- The formulation in the caption of Figure 1 "we introduce normalized mutual information" sounds a bit as if the current work has invented NMI. I would suggest changing the formulation. The same for "we introduce a new metric recall" in l.642.
- The formulation "To end it" in l. 424 sounds a bit off, perhaps "finally" is more fitting.
- I saw at least three different variants of capitalisation for "Scaled Lasso", "scaled lasso", and "scaled Lasso", and would suggest choosing one of them and sticking to it.
- Typo in l.386 "the t-the time point" should probably be "t-th time point".
- I believe a closing parenthesis that should be in l.402 has been placed in l.403.
- Missing word "be" in l.463 "This optimization problem can solved".
- L.610 refers to Eq. 33, which is in the appendix. I would suggest moving it to the main text in order to keep it self-contained.
- Type "selectd" in l.621
- Figure 3 is not referenced in the main text. This makes it hard for the reader to figure out when to look at it and what its message is.
- The text refers to the subfigures of Fig. 2, however, no subfigures are marked in Fig. 2.
- The sentence "the challenges of temporal independency assumptions, manual hyperparameter tuning, and the underutilization of multimodal data in dynamic network prediction" in the conclusion seems incomplete.

**Reviewer Confidence:**

3: The reviewer is confident but not certain that the evaluation is correct

**Scope:**

3: The work is somewhat relevant to the Web and to the track, and is of narrow interest to a sub-community

---

### Official Review · Reviewer_gAsR · 2024-12-04

**Novelty:** 5
**Technical Quality:** 5

**Review:**

The paper presents a novel, high-quality framework for dynamic dependence network (DDN) inference using multimodal data. It effectively addresses significant challenges in the domain, such as computational efficiency, prediction stability, and handling modality quality disparity, with a clear methodology and robust theoretical guarantees. The experimental validation across synthetic and real-world datasets demonstrates a thorough and reliable analysis.

pros:
1) The structure is well-organized, and technical concepts are explained with appropriate mathematical rigor.
2) Clear visualizations (e.g., time-varying graphs, dynamic functional connectivity) aid comprehension.

cons:
1) The heavy use of advanced mathematical concepts without simpler summaries might limit accessibility for a general audience.
2) Although the method is efficient, scalability in graphs with millions of nodes or higher dimensions is not fully explored.

**Questions:**

The paper mentions discretizing EEG data to reduce noise. How does the choice of discretization granularity affect the relationship matrix and overall inference accuracy?

**Reviewer Confidence:**

3: The reviewer is confident but not certain that the evaluation is correct

**Scope:**

3: The work is somewhat relevant to the Web and to the track, and is of narrow interest to a sub-community